# Three Stream Based Multi-level Event Contrastive Learning for Text-Video Event Extraction

**Jiaqi Li**[1,3]*, **Chuanyi Zhang**[2]*, **Miaozeng Du**[1,3], **Dehai Min**[1,3], **Yongrui Chen**[1,3], **Guilin Qi**[1,3]†

[1] School of Computer Science and Engineering, Southeast University, Nanjing, China
[2] College of Artificial Intelligence and Automation, Hohai University, Nanjing, China
[3] Key Laboratory of New Generation Artificial Intelligence Technology and Its
Interdisciplinary Applications (Southeast University), Ministry of Education, China
jqli@seu.edu.cn, 20231104@hhu.edu.cn, miaozengdu@seu.edu.cn
zhishanq@seu.edu.cn, yrchen@seu.edu.cn, gqi@seu.edu.cn

## Abstract

Text-video based multimodal event extraction refers to identifying event information from the given text-video pairs. Existing methods predominantly utilize video appearance features (VAF) and text sequence features (TSF) as input information. Some of them employ contrastive learning to align VAF with the event types extracted from TSF. However, they disregard the motion representations in videos and the optimization of contrastive objective could be misguided by the background noise from RGB frames. We observe that the same event triggers correspond to similar motion trajectories, which are hardly affected by the background noise. motivated by this, we propose a **T**hree **S**tream Multimodal **E**vent **E**xtraction framework (TSEE) that simultaneously utilizes the features of text sequence and video appearance, as well as the motion representations to enhance the event extraction capacity. Firstly, we extract the optical flow features (OFF) as motion representations from videos to incorporate with VAF and TSF. Then we introduce a Multi-level Event Contrastive Learning module to align the embedding space between OFF and event triggers, as well as between event triggers and types. Finally, a Dual Querying Text module is proposed to enhance the interaction between modalities. Experimental results show that TSEE outperforms the state-of-the-art methods, which demonstrates its superiority.

## 1 Introduction

Event extraction (EE) is a fundamental task which aims to recognize the event structure from texts (Nguyen et al., 2016; Nguyen and Grishman, 2015; Wadden et al., 2019; Lu et al., 2022). Recent years have witnessed the booming of the multimodal event extraction (MEE). MEE (Pratt et al., 2020; Sadhu et al., 2021; Li et al., 2017) extends EE

---

* J. Li and C. Zhang contributed equally to this work and should be considered co-first authors.
† Corresponding author.

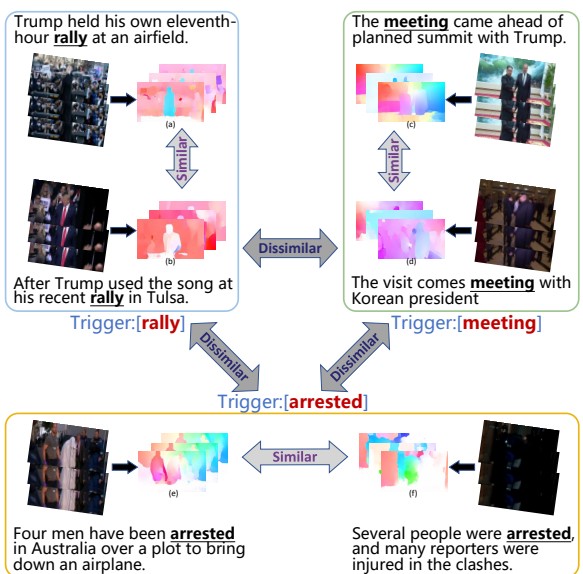

Figure 1: An illustration of the relationship between motion representations and event triggers. To capture the motion information, we extract optical flow features from each video. The optical flow features tend to be similar if the event triggers are the same. Otherwise, they are likely to become dissimilar.

by merging complementary information from multiple modalities such as texts, images or videos. Specifically, texts provide abstract semantics while visual data supplies concrete instances(Liu et al., 2023a,b; Yang et al., 2023a,b). Compared with text-image based MEE (TIMEE) (Li et al., 2022; Liu et al., 2022; Li et al., 2020; Zhang et al., 2017; Tong et al., 2020), text-video based MEE (TVMEE) (Chen et al., 2021; Wang et al., 2023) contains more context and scene information. Moreover, TVMEE presents temporal data that could capture the dynamic evolution of events, making it an area of significant interest.

Existing methods in TVMEE (Chen et al., 2021; Wang et al., 2023) extract text sequence features (TSF) and video appearance features (VAF) from texts and RGB frames by adopting pre-trained language and video models respectively. However,

they neglect the motion representations in videos. In TVMEE, motion representations may play an important role, as they furnish details on the motion and behavior of objects in videos. Furthermore, we observe that identical event triggers correspond to analogous motion representations. To explore the relationship between motion representations and event triggers, we introduce the optical flow features (OFF) (Dosovitskiy et al., 2015) as object motion representations. OFF (Ilg et al., 2017; Sun et al., 2018; Jiang and Learned-Miller, 2023; Marsal et al., 2023; Liu et al., 2021) represents the movement of objects in a sequence between consecutive frames and is extensively applied in video fields, such as video understanding (Teed and Deng, 2020; Luo et al., 2022), video super-resolution (Zheng et al., 2022; Chan et al., 2022), etc. As shown in Figure 1, we compare three triggers *'rally'*, *'meeting'* and *'arrested'*. For each sample we visualize the text, frames and OFF extracted from the corresponding frames. It could be observed that OFF is similar if it refers to the same trigger. In contrast, heterogeneous triggers usually point to dissimilar OFF.

A previous work (Wang et al., 2023) applies contrastive learning to reduce the distance between VAF and event types. Although VAF extracted from continuous frames may provide useful motion information, it also contains misguiding background noise. To be specific, the background noise is various scenes in heterogeneous videos. It does not provide any event semantics and can suppress the alignment between visual cues and event types. However, this issue could be properly alleviated by utilizing OFF because OFF solely exploits the object motion representations and filters out the scene information.

In this work, we design a novel framework, **T**hree **S**tream Multimodal **E**vent **E**xtraction (TSEE), which simultaneously leverages three modality features (text sequence, video appearance and motion representations) to improve the event extraction capability. To begin with, we employ pre-trained I3D (Carreira and Zisserman, 2017) and PWC (Sun et al., 2018) models to extract VAF and OFF from each video respectively. For the input text, we adopt a pre-trained language model (Devlin et al., 2018; Raffel et al., 2020) to obtain TSF. Then we propose a Multi-level Event Contrastive Learning (MECL) module, aiming to align the feature representations between OFF and event

triggers, as well as event types and triggers. We align each pair in the embedding space by introducing a multi-level contrastive objective. Lastly, we propose a Dual Querying Text (DQT) module to increase the interaction between modalities. In this module, VAF and OFF retrieve the cross-modality attention of each token in TSF respectively.

The contributions of our work could be summarized as follows:

• We propose a novel framework called TSEE that leverages the motion representations in videos. To the best of our knowledge, we are the first to introduce optical flow features into TVMEE.

• Our proposed modules, MECL and DQT, significantly improve the model performance. MECL aligns the embedding space of OFF, event triggers and types. DQT enhances the interaction among text, video and optical flow modalities.

• The experimental results on two benchmark datasets demonstrate the superiority of our framework over the state-of-the-art methods.

## 2 Related Work

### 2.1 Event Extraction

In the field of event extraction research, the initial work primarily focused on sentence-level studies in the text. Some works have explored the use of convolutional neural networks (Nguyen and Grishman, 2015; Nguyen et al., 2016), recurrent neural networks (Nguyen and Grishman, 2015; Liu et al., 2019, 2020), graph neural networks (Li et al., 2017), and later emerging pre-trained language models (Wadden et al., 2019; Wang et al., 2022; Lu et al., 2022) for handling the extraction of triggers and arguments. In the field of computer vision, event extraction is operationalized as situation recognition (Pratt et al., 2020; Sadhu et al., 2021; Yatskar et al., 2016; Li et al., 2017), with tasks primarily involving the classification and extraction of frames containing entities and roles (arguments) from images with actions (visual events)(Zhang et al., 2021; Chen et al., 2022a,b). In recent years, there has been an emergence of using multimodal information for event extraction(Chen et al., 2022c, 2023). (Zhang et al., 2017) demonstrated the effectiveness of using visually based entity data to extract events. Previous multimodal event extraction models (Li et al., 2020; Liu et al., 2022) mostly dealt with visual data in the form of images, (Chen et al., 2021) pioneered a model that can jointly extract events from text and video data. They used

a pre-trained text-video retrieval model to find the most relevant text-video pairs. Based on (Chen et al., 2021)'s approach, (Wang et al., 2023) introduced supervised contrastive learning to enhance the representation of the two modalities for further event extraction.

## 2.2 Supervised Contrastive Learning

Contrast learning is a technique that trains models to distinguish between similar and different examples. Self-supervised representation learning methods such as (Kim et al., 2020; Yue et al., 2022; Kim et al., 2021; Kaku et al., 2021; Iter et al., 2020) divide each sample into positive and negative samples, learning feature representations by comparing the similarity between sample pairs. Works such as (Gunel et al., 2020; Wu et al., 2022; Gunel et al., 2020; Song et al., 2022) optimize the supervised contrastive objective for supervised contrastive learning. For event extraction tasks, (Wang et al., 2021) proposes a contrastive pre-training framework that uses semantic structures. (Yao et al., 2022) introduces an efficient event extraction model with a contrastive objective to distinguish triggers and arguments. (Zolfaghari et al., 2021) presents a more effective cross-modal contrastive learning loss function, compared to directly using loss functions designed for visual data.

## 2.3 Optical Flow

Most of the existing methods for extracting optical flow rely on pixel-by-pixel prediction using neural networks. Optical flow extraction models have various model structures, including encoder-decoder architecture (Dosovitskiy et al., 2015), iterative refinement of multiple FlowNet modules using cumulative superposition (Ilg et al., 2017), feature processing and extraction through a pyramid structure (Sun et al., 2018), and construction of cost volumes with different expansion coefficients (Jiang and Learned-Miller, 2023). (Marsal et al., 2023) trains optical flow estimation networks by training two networks that jointly estimate optical flow and brightness changes. (Liu et al., 2021) addresses the optical flow problem of occluded pixels. (Chan et al., 2022) utilizes temporal information and proposes a novel approach to combine optical flow and deformable alignment in videos. (Huang et al., 2019) employs optical flow to solve motion detection issues related to dynamic background and changing foreground appearance.

## 3 Approach

### 3.1 Task Definition

Given a text-video pair $(T, V)$, we denote the sequence of input text tokens as $T = \{t_1, t_2, ..., t_m\}$. We sample from the video every 16 frames to get the clip sequence $V = \{c_1, c_2, ..., c_k\}$. In TVMEE, each sample is annotated a set of event types $E = \{e_1, e_2, ...\}$. Our goal is to jointly extract event triggers and event arguments. An event trigger is a word or phrase in a sentence that indicates the occurrence of an event. For example, in the sentence '*John **bought** a new car yesterday*', the word '*bought*' is the event trigger, indicating the occurrence of a buying event. Event argument extraction is to identify relevant pieces of information or arguments from texts. The pieces commonly involve an event, such as subject, object, verb, and other modifiers. Then these roles are mapped to their semantic roles such as agent, patient, location, time, and so on. Take the above sentence as an example, '*John*', '*car*' and '*yesterday*' are the event arguments referring to the buying event and the roles are '*buyer*', '*product*' and '*time*' respectively.

### 3.2 Feature Extraction

Our framework utilizes information from both text and video features as shown in Figure 2. In particular, the video incorporates features from two perspectives. The first is the video appearance features, which represents color, texture, shape, and other visual cues. Secondly, motion features provide information about dynamics of objects within the scene. We employ corresponding pre-trained models to extract these features respectively.

**Text feature extraction.** The input text tokens are encoded using pre-trained T5-base (Raffel et al., 2020) with $d_t$ hidden dimensions. Thus each input sequence is represented as a matrix $F_T \in \mathbb{R}^{n_l \times d_t}$, where $n_l$ is the length of sequence.

**Video feature extraction.** We input each clip sequence into the I3D network pretrained on Kinetics dataset and the PWC network pretrained on Sintel dataset. Then we obtain a sequence of VAF and OFF. To represent a video-level feature, we sum up all the features within the sequence . VAF and OFF are denoted as $F_V \in \mathbb{R}^{d_v}$ and $F_O \in \mathbb{R}^{d_o}$.

### 3.3 Multi-level Event Contrastive Learning

We observe that identical event triggers usually involve similar motion representations, which are not affected by background noise. Additionally, in

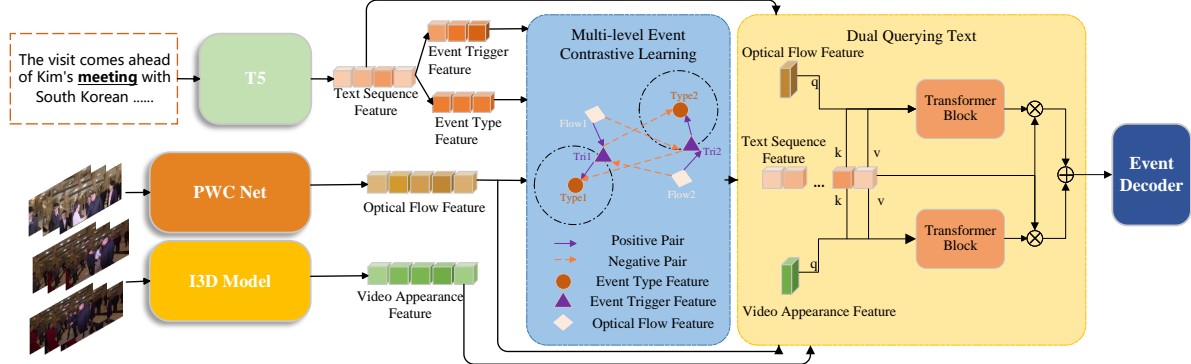

Figure 2: The overview of TSEE framework. We leverage a T5 model to extract Text Sequence Feature (TSF). For each video we adopt a PWC Net and an I3D model to obtain Optical Flow Feature (OFF) and Video Appearance Feature (VAF) respectively. Event Trigger Feature and Event Type Feature are extracted from TSF. Then the two features along with OFF are entered into a Multi-level Event Contrastive Learning module to align the feature representations between them. After that, Dual Querying Text module enhances the interaction among TSF, VAF and OFF. Finally, the event decoder takes the aggregated feature as input to predict events.

the event extraction, an event type is correlated to various triggers. Motivated by the above observations, we propose a Multi-level Event Contrastive Learning (MECL) module. This module aligns the feature representations between OFF and triggers. The embedding spaces of event types and triggers are also aligned using this module. We apply supervised contrastive learning (Khosla et al., 2020) in this module and define multi-level labels for different event levels.

**Event type level.** Since an event type corresponds to various triggers, we use event types as the anchors for triggers. Our purpose is to push the triggers referring to the identical event type close. In this level, we define all the event types of the dataset event schema as the label set $E = \{e_1, e_2, ..., e_p\}$, where $p$ is the number of event types in the dataset event schema.

**Event trigger level.** Considering the same event triggers correspond to similar motion trajectories in videos, we regard the triggers as the anchors for OFF. The label set in this level is all the triggers $W = \{w_1, w_2, ...\}$ in the dataset. For each trigger we could obtain the embedding index from pretrained language model as the label index.

Given a batch of $N$ samples, we first select the samples annotated with one event type for computing contrastive loss. It is for the reason that if a sample has more than one event, OFF may contain multiple motion trajectories. Thus OFF could not be directly assigned the certain single label of event. After filtering the samples, we obtain a smaller batch of OFF $F_{O_c}$, the trigger words $W_c$, as

well as the corresponding event types $E_c$. For the Event type level, positive pairs of each event type consist of all referring trigger words and the event type itself. In contrast, the negative pairs comprise irrelevant trigger words and the event type itself. For Event trigger level, each trigger's positive pairs are composed of optical flow features that point to the trigger and the trigger. Conversely, the negative pairs are made up of optical flow features that are unrelated to the trigger and the trigger itself.

Considering the $i$-th sample in this smaller batch, we first enter $w^i$ and $e^i$ into a pre-trained T5-base model to obtain respective feature representations:

$$z^i = T5(w^i),$$
$$x^i = T5(e^i). \qquad (1)$$

Then we adopt the supervised contrastive learning to optimize contrastive loss of the Event type level and Event trigger level $\mathcal{L}_{type}$ and $\mathcal{L}_{trig}$:

$$\mathcal{L}_{type} = -\sum_{i=1}^{B} log \frac{exp(x^i \cdot z^i / \tau)}{\sum_{z^l \in W_c \setminus z^i} exp(x^i \cdot z^l / \tau)},$$
$$\mathcal{L}_{trig} = -\sum_{i=1}^{B} log \frac{exp(z^i \cdot F_O^i / \tau)}{\sum_{F_O^u \in F_{O_c} \setminus F_O^i} exp(z^i \cdot F_O^u / \tau)}, \qquad (2)$$

where $B$ is the number of samples after filtering, and $\tau$ is the temperature parameter of supervised contrastive learning. Finally the multi-level loss $\mathcal{L}_{multi}$ is defined as :

$$\mathcal{L}_{multi} = \mathcal{L}_{type} + \mathcal{L}_{trig}. \qquad (3)$$

Formally, the Multi-level Event Contrastive Learning algorithm is shown as Algorithm 1.

### 3.4 Dual Querying Text

We design a Dual Querying Text (DQT) module to enhance the interaction among three modalities. The intuition is to query TSF which token responds to VAF or OFF. For example, if the input text has the word *Police* describing the argument of the event, this token would respond to VAF. It is because VAF may contain visual cues whose semantics are close to this argument. To encode the dual queries of TSF, We utilize two transformer architectures. The attention scores of each token reflect the degree of response to VAF or OFF.

For TSF, VAF, OFF denoted as $F_T \in \mathbb{R}^{n_l \times d_t}$, $F_V \in \mathbb{R}^{d_v}$ and $F_O \in \mathbb{R}^{d_o}$ in Section 3.2, VAF and OFF are projected into queries respectively. In both transformer architectures, TSF is projected to obtain keys and values. Then we adopt a softmax function to calculate the dual attention weights:

$$A_v = softmax(\frac{F_V H_{q_1} H_{k_1}^\top F_T^\top}{\sqrt{dt}})F_T H_{v_1},$$

$$A_o = softmax(\frac{F_O H_{q_2} H_{k_2}^\top F_T^\top}{\sqrt{dt}})F_T H_{v_2}, \quad (4)$$

Where $H_q, H_k, H_v$ are three projection matrices for query, key and value respectively. The output attention scores are aggregated as follows:

$$F_A = A_v \cdot F_T + A_o \cdot F_T \quad (5)$$

## 4  Experiment

**Datasets.** We evaluate our approach on two open-ended TVMEE datasets: TVEE (Wang et al., 2023) and VM2E2 (Chen et al., 2021). **TVEE** dataset contains 7598 text-video pairs. The international news videos with captions are collected from the On Demand News channel. The event schema is from the ACE2005 (Walker, 2006) benchmark that consists of 8 superior event types and 33 event types. *Contact.Speech*, *Disaster.Disaster* and *Accident.Accident* are added to the event schema because the schema in ACE2005 could not cover all the event types in videos. The TVEE dataset is randomly divided into train, valid, and test sets in a ratio of 8:1:1. **VM2E2** is a collection of text and video data that includes 13,239 sentences and 860 videos. Within the dataset, there are 562 pairs of sentences and videos that share the same event

---

**Algorithm 1** Multi-level Event Contrastive Learning

**Require:** OFF $F_O$, event types $E = \{e_1, e_2, ...\}$, event triggers $W = \{w_1, w_2, ...\}$, event type positive pairs $S_{py} = \emptyset$, event type negative pairs $S_{ny} = \emptyset$, event trigger positive pairs $S_{pg} = \emptyset$, event trigger negative pairs $S_{ng} = \emptyset$, filtering batch $S_f = \emptyset$, supervised contrastive learning function CON.

1: **for** $(F_O, E)^i$ in batch **do**
2:   **if** Len($E^i$)==1 **then**
3:     $z^i \leftarrow T5(w^i)$
4:     $x^i \leftarrow T5(e^i)$
5:     $S_f$.append($z^i, x^i, F_O^i$)
6:   **else**
7:     CONTINUE
8:   **end if**
9: **end for**
10: **for** $(z, x, F_O)^j$ in $S_f$ **do**
11:   **if** $z^j$ refers to $x^j$ **then**
12:     $S_{py}$.append($z^j, x^j$)
13:   **else**
14:     $S_{ny}$.append($z^j, x^j$)
15:   **end if**
16:   **if** $F_O^j$ refers to $z^j$ **then**
17:     $S_{pg}$.append($z^j, F_O^j$)
18:   **else**
19:     $S_{ng}$.append($z^j, F_O^j$)
20:   **end if**
21: **end for**
22: $\mathcal{L}_{type} = \text{CON}(S_{py})+\text{CON}(S_{ny})$
23: $\mathcal{L}_{trig} = \text{CON}(S_{pg})+\text{CON}(S_{ng})$
24: **return** $\mathcal{L}_{type} + \mathcal{L}_{trig}$

---

type, with each pair containing only one event. The dataset defines 16 multimodal event types based on the LDC ontology. Following (Chen et al., 2021), we split VM2E2 into 411 and 151 samples.

**Evaluation Metrics.** Following (Wang et al., 2023), we utilize the same evaluation metrics to report text, video and multimodal evaluation results. The evaluation metrics include: Precision (P), Recall (R) and F-score (F1). The performance of text event extraction is evaluated by two subtasks: event trigger extraction and event argument extraction. The correctness of a trigger prediction is determined by whether its type and span align with the labels, while for an argument prediction, it is determined by whether its span and all the roles

| Dataset | Input | Model | Text Evaluation | | | | | | Video Evaluation | | | Multimodal Evaluation | | |
|---|---|---|---|---|---|---|---|---|---|---|---|---|---|---|
| | | | Trigger | | | Argument | | | | | | | | |
| | | | P | R | F1 | P | R | F1 | P | R | F1 | P | R | F1 |
| **TVEE** | **Text** | DEEPSTRUCT | 76.4 | 75.2 | 75.8 | 53.1 | 48.9 | 50.9 | - | - | - | 76.4 | 75.2 | 75.8 |
| | | CoCoEE$_T$ | 76.0 | 76.6 | 76.3 | 62.9 | 44.2 | 51.9 | - | - | - | 76.0 | 76.6 | 76.3 |
| | | TSEE$_T$ | 75.7 | 77.2 | 76.4 | 63.3 | 45.0 | 52.6 | - | - | - | 75.7 | 77.2 | 76.4 |
| | **Video** | JSL | - | - | - | - | - | - | 48.2 | 51.6 | 49.8 | 48.2 | 51.6 | 49.8 |
| | | CoCoEE$_V$ | - | - | - | - | - | - | 49.1 | 60.7 | 54.3 | 49.1 | 60.7 | 54.3 |
| | | TSEE$_V$ | - | - | - | - | - | - | 48.7 | 62.1 | 54.6 | 48.7 | 62.1 | 54.6 |
| | **Multimodal** | JMMT | 74.3 | 80.2 | 77.1 | 50.1 | **54.9** | 52.3 | 55.4 | 57.0 | 56.2 | 87.2 | 88.6 | 87.9 |
| | | CoCoEE | 80.7 | 76.4 | 78.5 | 65.6 | 45.4 | 53.6 | 56.4 | 57.4 | 56.9 | 92.9 | 92.9 | 92.9 |
| | | **TSEE (ours)** | **82.6** | **80.5** | **81.5** | **67.0** | 49.3 | **56.8** | **58.2** | **58.6** | **58.4** | **94.4** | **93.7** | **94.0** |
| **VM2E2** | **Text** | DEEPSTRUCT | 44.7 | 43.1 | 43.9 | 19.8 | 13.2 | 15.9 | - | - | - | 44.7 | 43.1 | 43.9 |
| | | CoCoEE$_T$ | 41.5 | 45.6 | 43.5 | 20.5 | 15.3 | 17.5 | - | - | - | 41.5 | 45.6 | 43.5 |
| | | TSEE$_T$ | 45.2 | 41.8 | 43.4 | 21.2 | 17.1 | 18.9 | - | - | - | 45.2 | 41.8 | 43.4 |
| | **Video** | JSL | - | - | - | - | - | - | 21.2 | 18.6 | 19.8 | 21.2 | 18.6 | 19.8 |
| | | CoCoEE$_V$ | - | - | - | - | - | - | 27.3 | 31.2 | 29.1 | 27.3 | 31.2 | 29.1 |
| | | TSEE$_V$ | - | - | - | - | - | - | 26.5 | 30.7 | 28.4 | 26.5 | 30.7 | 28.4 |
| | **Multimodal** | JMMT | 39.7 | **56.3** | 46.6 | 17.9 | 24.3 | 20.6 | 32.4 | 37.5 | 34.8 | 76.1 | 69.5 | 72.7 |
| | | CoCoEE | 47.3 | 47.7 | 47.5 | **26.7** | 18.5 | 21.8 | 33.2 | 37.2 | 35.1 | 78.2 | 75.6 | 76.9 |
| | | **TSEE (ours)** | **49.2** | 53.5 | **51.6** | 24.5 | **27.4** | **25.9** | **35.1** | **38.0** | **36.5** | **78.9** | **77.2** | **78.0** |

Table 1: Comparison with the state-of-the-art methods. The evaluation metrics are introduced in Section 4.1. The best performed methods in each metric are highlighted in bold.

align with the labels.

**Implementation Details.** We use Pytorch and a 2080 Ti GPU to implement our framework and conduct experiments. We apply a pre-trained T5-base (Raffel et al., 2020), as the TSF encoder. For the video input, we separately adopt pre-trained I3d (Carreira and Zisserman, 2017) and PWC (Sun et al., 2018) to extract VAF and OFF. For the event extraction decoder, we use CRF decoder following (Wang et al., 2023). The dimension of TSF, VAF and OFF are 768, 1024 and 1024 respectively. We utilize a linear function to project the dimension of VAF and OFF to 768. Following (Wang et al., 2023), we train our model for 15 epochs and the batchsize is set 16. The optimizer is Adam and the learning rate is 10e-5. Following (Yao et al., 2022), we utilize 0.3 for the parameter $\tau$ in MECL.

### 4.1 Baselines

Following (Wang et al., 2023), we compare our model with other methods in three settings, which are **Text Event Extraction**, **Video Event Extraction**, **Multimodal Event Extraction**.
**Text Event Extraction.** For text event extraction, we only utilize text input. We compare the following models in this setting:
- **DEEPSTRUCT** (Wang et al., 2022) : It is the state-of-the-art method in text event extraction.

It proposes structure pretraining to let language model understand the structure in the text.
-**CoCoEE$_T$** (Wang et al., 2023) : It uses the text encoder and a CRF decoder of CoCoEE without CoLearner module.
-**TSEE$_T$** : It utilizes the T5-base encoder and a CRF encoder to extract events with text modality. It is without MECL module and DQT module.
**Video Event Extraction.** We only use video input as the video event extraction. We compare the models as follows:
-**JSL (Pratt et al., 2020)** :We follow (Wang et al., 2023) to use a sota model JSL in video event extraction. Key frames are utilized to detect events.
-**CoCoEE$_V$** (Wang et al., 2023) : It utilizes the video encoder of CoCoEE and a video event decoder without CoLearner module.
-**TSEE$_V$** : It utilizes a pre-trained I3D model to extract video features and the decoder is set the same as (Wang et al., 2023). It is also without MECL module and DQT module.
**Multimodal Event Extraction.** This is our full task setting. We compare the models as follows:
-**JMMT (Chen et al., 2021)** :It utilizes a transformer encoder to jointly encode the text and video inputs. The visual features include video-level features and image-level features.
-**CoCoEE (Wang et al., 2023)** :It is the state-of-the-art model in text-video based event extraction.

| Dataset | Units | | | Trigger | | | Argument | | |
|---|---|---|---|---|---|---|---|---|---|
| | O | H | D | P | R | F1 | P | R | F1 |
| TVEE | | | | 76.2 | 76.9 | 76.5 | 62.8 | 46.1 | 53.2 |
| | ✔ | | | 76.8 | 77.3 | 77.0 | 63.9 | 45.7 | 53.3 |
| | ✔ | ✔ | | 80.5 | 79.2 | 79.8 | 64.5 | 47.3 | 54.6 |
| | ✔ | ✔ | ✔ | **82.6** | **80.5** | **81.5** | **67.0** | **49.3** | **56.8** |
| VM2E2 | | | | 42.3 | 45.9 | 44.0 | 21.3 | 16.6 | 18.7 |
| | ✔ | | | 44.0 | 47.2 | 45.5 | 20.8 | 18.1 | 19.4 |
| | ✔ | ✔ | | 47.9 | 50.6 | 49.2 | 22.7 | 25.3 | 23.9 |
| | ✔ | ✔ | ✔ | **49.2** | **53.5** | 51.6 | **24.5** | **27.4** | **25.9** |

Table 2: Ablation study on three units in TSEE. 'O' represents OFF (Optical Flow Features). 'H' means MECL (Multi-level Event Contrastive Learning) module. 'D' denotes DQT (Dual Querying Text) module. '✔' represents our framework is equipped with the unit.

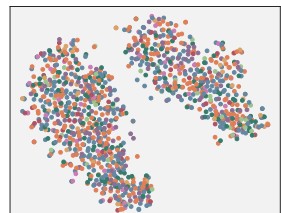 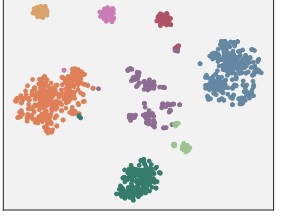

(a) w/o MECL      (b) w/ MECL

Figure 3: T-SNE visualization for MECL module. **w/o MECL** (a) denotes that MECL is removed from TSEE and **w/ MECL** (b) means that TSEE is trained with MECL module. Each dot represents one OFF and each color denotes a specific event trigger.

It contrasts the event types and video features.

## 4.2 Main Results

The experiment results on TVEE and VM2E2 datasets are presented in Table 1. We could find that when the input data only consists of text, DEEPSTRUCT, CoCoEE$_T$ and TSEE$_T$ achieve similar performance. Specifically, on TVEE dataset TSEE$_T$ performs 0.1% and 0.6% better than CoCoEE$_T$ on F1 of trigger extraction. However, DEEPSTRUCT reaches a higher F1 than TSEE$_T$ and CoCoEE$_T$ on VM2E2 dataset. This could be because the ability to extract event information from text of the three models is comparable.

When there is only video data in input, CoCoEE$_V$ and TSEE$_V$ show the comparable performance and are both better that JSL on two datasets. It is because JSL applies to static frames while CoCoEE$_V$ and TSEE$_V$ adopt pre-trained models of videos to capture dynamic information.

The results of multimodal data input show that TSEE achieve the best performance on most of the evaluation metrics compared with existing state-of-the-art methods. On TVEE dataset, our model reaches 81.5% F1 of trigger extraction compared with 77.1% and 78.5% achieved by JMMT and Co-CoEE respectively. This result demonstrates that the integration of motion representations in videos is helpful for the multimodal event extraction task. On VM2E2 dataset, the F1 score of trigger extraction is improved from 47.5% (CoCoEE) to 51.7%, where the improvement is larger than that of TVEE dataset. This may be the reason that in VM2E2 dataset, each sample is annotated with only one event. The MECL module would not filter any sample in every batch when computing contrastive loss, thus obtaining better feature representations and boosting the performance of the model. We notice that JMMT performs well in the recall metric, such as 54.9% argument extraction recall on TVEE dataset and 56.3% trigger extraction recall on VM2E2 dataset. This may be that JMMT utilizes the additional object detection model to inject proposal features of key frames to the transformer encoder, improving the recall of triggers and arguments in samples.

We also observe that the results show the similar trends from single modality to multimodal input, which verifies that injecting multimodal input to TSEE and CoCoEE both boosts the performance in all metrics. Specifically, the incorporation of video to TSEE$_V$ boosts the F1 performance from 76.4% to 81.5% on TVEE dataset. For CoCoEE$_V$, the F1 score is improved from 76.3% to 78.5%.

## 4.3 Ablation Study

To validate the effectiveness of different innovations and parts in TSEE, we conduct ablation studies on the two datasets. We investigated three main units of TSEE: (1) Integration of optical flow feature; (2) Multi-level Event Contrastive Learning module; (3) Dual Querying Text module. The baseline in the first line applies sum function to VAF and TSF. The results are summarized in Table 2.

**Effectiveness of OFF.** In this part we extract OFF from video data and sum up VAF, TSF and OFF. From Table 2, we observe that the integration of OFF improves all evaluation metrics over baseline on the two datasets, verifying that OFF provides beneficial information for event extraction.

**Effectiveness of MECL.** To evaluate the influence of MECL module, we utilize MECL module based on the second line of each part. As shown in Table 2, MECL module brings the most improvement to our framework, such as 2.8% trigger extraction F1

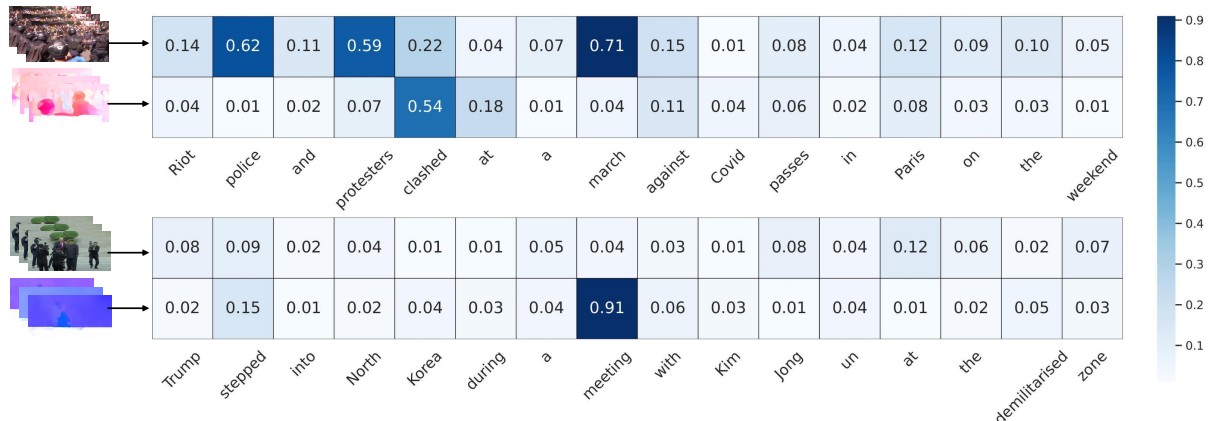

Figure 4: Case study on DQT module. Attention heatmaps of two randomly sampled examples are visualized. In each example, the first line is VAF and the second line is OFF.

score on TVEE dataset and 3.7% trigger extraction F1 score on VM2E2 dataset. This demonstrates that MECL module could refine the feature representations and align the features between heterogeneous modalities, thus boosting the performance.

**Effectiveness of DQT.** We also evaluate the impact of DQT module. From Table 2, we could find that DQT module improves the performance of all evaluation metrics significantly on the two datasets. It is worth noting that the recall metric of trigger extraction is boosted from 50.6% to 53.5% on VM2E2 dataset and so is argument extraction recall metric from 25.3% to 27.4%. The reason is perhaps that in the DQT module, each text token is queried by the VAF and OFF, thus enhancing the ability of searching instances contained in videos.

### 4.4 Visualization of T-SNE for MECL

To verify the impact of MECL module, we use t-SNE (Van der Maaten and Hinton, 2008) to visualize the manifold of TSEE with and without MECL module. Our MECL module is designed to reduce the distance between OFF and event triggers. We randomly sampled 1500 OFF trained with or without MECL module on TVEE dataset. The visualization results are shown in Figure 3, where OFF belonging to the same trigger is marked in the same color. It could be clearly seen that OFF trained with MECL module in subfigure is obviously separated into various compact clusters. However, when OFF is not trained with MECL module, there is no distinctiveness between OFF belonging to different triggers. This result demonstrates that our MECL module does well in aligning the semantics of different modalities.

### 4.5 Case Study on DQT

In order to intuitively show the effectiveness of DQT module, we conduct case studies on TVEE dataset. As shown in Figure 4, we visualize the attention heatmaps based on the attention scores output by DQT. As DQT utilizes VAF and OFF to query each token in TSF respectively, each sample corresponds to two lists of attention scores. From Figure 4, we could observe that for each sample, the frame appearance or motion related tokens are paid more attention by VAF or OFF. In the first example, When VAF queries *police*, *protesters* and *march*, it gives more attention scores than other tokens. We could also observe that OFF attends to the *clashed* most. In the second example, each token is allocated similar attention score by VAF. This may be the reason that the pre-trained I3D model does not have the knowledge of instances such as *Trump* and *Kim*. OFF gives a higher attention score to *meeting* because this token could provide motion information. From the above analysis we can see that DQT module does well in understanding the relationship between multimodal semantics.

### 5 Conclusion

In this paper, we propose a Three Stream Multimodal Event Extraction framework that explores the utilization of motion representations in text-video based multimodal event extraction (TVMEE) tasks. Optical flow features are extracted from videos as motion representations to incorporate with other modalities. To improve alignment among feature representations, we propose a Multi-level Event Contrastive Learning module. A Dual Querying Text module is also designed to help en-

hance the interaction between different modalities. TSEE achieves the state-of-the-art results on two datasets, demonstrating the effectiveness of our framework. In future work, we will explore the utilization of large language model (LLM) in fusing modality features to boost TIMEE performance.

## Limitations

The main limitation of our work is the offline training. As the insufficiency of GPU resources, we need to extract the VAF and OFF in advance and could not optimize the video pre-trained model online. The other limitation is the inapplicability of open-domain event extraction. As both two datasets are annotated in a close-domain event set, our framework can not deal with open-domain event extraction.

## Acknowledgements

This work is partially supported by National Nature Science Foundation of China under No. U21A20488, 62302149 and 62372155. We thank the Big Data Computing Center of Southeast University for providing the facility support on the numerical calculations in this paper.

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
