# OpenReview forum: "Three Stream Based Multi-level Event Contrastive Learning for Text-Video Event Extraction"
_EMNLP/2023/Conference — EMNLP 2023 Main_

### Official Review · Reviewer_ZJRe · 2023-08-05

**Soundness:** 4

**Excitement:**

3: Ambivalent: It has merits (e.g., it reports state-of-the-art results, the idea is nice), but there are key weaknesses (e.g., it describes incremental work), and it can significantly benefit from another round of revision. However, I won't object to accepting it if my co-reviewers champion it.

**Paper Topic And Main Contributions:**

The paper proposes a new approach in text-video based multimodal event extraction. They introduce three components - 1) utilizing the optical features for this task, 2) multi-level event contrastive learning to align embedding spaces, and 3) dual querying text which maps the tokens to visual appearance feature (VAF) or optical flow feature (OFF). The authors show that the approach outperforms unimodal and multimodal baselines and show that majority of the improvement comes from components 2 and 3.

**Reasons To Accept:**

1. The proposed approach is logical. Given the event extraction task, utilizing flow based feature is intuitive and utilizing a query module to map tokens to this as well as using contrastive learning based on event types and triggers are well founded.
2. The experimental results are solid with clarity on the effect of each component.
3. The paper is well written with clear explanation of the approach and the experimental setup making the results reproducible.

**Reasons To Reject:**

1. Optical flow feature is commonly used in vision and calling using this as first novelty lacks basis.
2. Section 3.1 of task definition is not well explained. Authors start by mention the setting of multimodal event extraction but only give an example of text based event.
3. The authors mention in the abstract that previous research disregard motion representation in videos. This is not correct. Video transformers do capture motion.

**Reproducibility:**

4: Could mostly reproduce the results, but there may be some variation because of sample variance or minor variations in their interpretation of the protocol or method.

**Reviewer Confidence:**

4: Quite sure. I tried to check the important points carefully. It's unlikely, though conceivable, that I missed something that should affect my ratings.

---

> ### Author Rebuttal · Authors · 2023-08-26
>
> Thank you for your critical feedback and suggestions. We address your thoughts point by point below.
>
> > **Q1**: Optical flow feature is commonly used in vision and calling using this as first novelty lacks basis.
>
> **A1**: Thank you for your thoughtful question. Our statement on the contribution may not be appropriate and it may cause misunderstanding. In TVMEE subtask, we are the first to associate the optical flow feature with the event triggers. This correlation is hardly influenced by background noise. Moreover, the performance is improved by our proposed contrastive learning module which aligns the embedding space of two elements. Thus, we regard introducing the optical flow feature as an important contribution and a good novelty. We agree with you that just mentioning optical flow feature may not be the first novelty and contribution. We will restate our first novelty in the final revised version.
>
> > **Q2**: Section 3.1 of task definition is not well explained.
>
> **A2**: Thank you for your rigorous review. Our task is to identify the triggers and arguments in text. For video modality, there is no trigger and argument annotation. We represent events in videos using event type as the definition of line 212 $E=(e_{1},e_{2},...) $.  We will add the explanation in our final version.
>
> > **Q3**: The authors mention in the abstract that previous research disregard motion representation in videos. This is not correct.
>
> **A3**: Thank you for your pertinent comment. What we claim in the abstract may cause ambiguity. There are indeed many works in other related fields of video that utilize motion information. However, previous research we mentioned refers to works on Text-Video event extraction.  They ignored the motion representations as illustrated in the introduction section. Nevertheless, we observed the correlation between motion representations and event triggers and then introduced this correlation to our framework.  We will make our statement clearer in our final version.

---

### Official Review · Reviewer_aGn9 · 2023-08-05

**Typos Grammar Style And Presentation Improvements:** in Fig 3. I believe there is a typo! …
**Soundness:** 3

**Excitement:**

3: Ambivalent: It has merits (e.g., it reports state-of-the-art results, the idea is nice), but there are key weaknesses (e.g., it describes incremental work), and it can significantly benefit from another round of revision. However, I won't object to accepting it if my co-reviewers champion it.

**Paper Topic And Main Contributions:**

In this frame, The authors leverage three modality features text sequence, video, and motion representations. The observation and motivation for the problem is clear. The authors utilize contrastive loss between optical flow features and even triggers as well as event triggers and event types to make sure similar motion representations correspond to similar event triggers. This way the framework extracts the relevant information from the video required to generate a suitable event trigger and type. Further studies show this framework generates state-of-the-art results in various evaluation metrics. Their claims are also supported by ablation studies.

**Questions For The Authors:**

contrastive learning usually requires a lot of data for the negative and positive pairs formation, since the existing dataset is small it may or may not affect the result as you scale up. Is there any way you can think about any idea borrowed from score based/diffusion models for this framework? It might be helpful in a dual way from image to event trigger and also event trigger to image generation as it may learn the joint distribution of the embeddings.

**Reasons To Accept:**

1. observation is good. Using extra visual features and fusing them into the learning is a good way of utilizing the visual feature more to understand the event.
2. T-SNE visualization looks promising in the ablation study part, the clusters support the claimed results and observation.
3. The proposed method outperforms almost all the parameters in the evaluation section.

**Reasons To Reject:**

1. Is there any related work that supports the claim of affecting event trigger extraction from videos due to the other noise in the frames?

2. "We sample from the video every 16 frames to get the clip sequence"- is there any specific reason to sample 16 frames? did you experiment with any other sampling algorithm i.e. importance sampling? what if for each epoch you randomly sample frames and feed it?

3.  is it possible to replace the optical flow feature and use the saliency feature from the video frames, Although saliency does not represent motion information it can help with suppressing the background noise and only focus on the relevant information required for the event trigger.



**Reproducibility:**

4: Could mostly reproduce the results, but there may be some variation because of sample variance or minor variations in their interpretation of the protocol or method.

**Reviewer Confidence:**

4: Quite sure. I tried to check the important points carefully. It's unlikely, though conceivable, that I missed something that should affect my ratings.

---

> ### Author Rebuttal · Authors · 2023-08-26
>
> We are grateful for your attentive comments and providing thoughtful feedback on our work. We will provide our insights point by point below.
>
> > **Q1**: Is there any related work that supports the claim of affecting event trigger extraction from videos due to the other noise in the frames?
>
> **A1**: Thank you for your thoughtful question. Works [1], [2] and [3] could all support the claim that the background noise in the frames affects event trigger extraction. In their works, they directly propose to obviate the influence of background noise of the frames or images for better information extraction. We will add the related work to support the claim in our final revised version.
>
> > **Q2.1**: Specific reason to sample 16 frames.
>
> **A2.1**: Thank you for your valuable question.  We sample from the video every 16 frames because the input for backbones ( I3D Model and PWC Net ) are restricted to 16 frames. In fact, most backbones ( [4], [5] ) for video features extraction are required 16 frames input such as C3D, R3D and S3D.
>
> > **Q2.2**: Other sampling algorithm such as importance sampling and random sampling.
>
> **A2.2**: Thank you for your pertinent suggestion. We suppose that importance sampling could provide another insight in extracting video appearance features. For video appearance features which mainly provide the scene information, it does not really need to sample continuous frames. To validate the idea, we conduct several additional experiments. We utilize the key frame sampling algorithm of OpenCV as the importance sampling strategy.  Besides, we conduct the experiment of random sampling. The results are summarized as follows:
>
>
>
> |        Model        | Trigger F1 | Argument F1 | Video F1 | Multimodal F1 |
> | :-----------------: | :--------: | :---------: | :------: | :-----------: |
> | Importance sampling |    80.2    |    55.1     |   56.2   |     92.4      |
> |   Random sampling   |    79.3    |    53.7     |   55.9   |     90.7      |
> |     TSEE (ours)     |    81.5    |    56.8     |   58.4   |     94.0      |
>
> The results show that changing sampling strategy does not help improve the performance for our framework.
>
>
>
> > **Q3**: replace the optical flow feature and use the saliency feature from the video frames
>
> **A3**: Thank you for your critical consideration. Both saliency feature and optical flow feature could alleviate the background noise. We utilize ITSD [6] to extract salience feature. The experiment results are shown as follows:
>
>
>
> |       Model       | Trigger F1 | Argument F1 | Video F1 | Multimodal F1 |
> | :---------------: | :--------: | :---------: | :------: | :-----------: |
> | TSEE (two stream) |    76.5    |    53.2     |   52.5   |     89.9      |
> | Salience feature  |    79.8    |    55.4     |   56.7   |     91.3      |
> |    TSEE (ours)    |    81.5    |    56.8     |   58.4   |     94.0      |
>
> TSEE (Two stream) means only VAF (video appearance feature) and TSF (text sequence feature) are fed into the framework. It could be observed that incorporating salience feature achieves higher performance than TSEE (two stream). This result demonstrates the effectiveness of sailence feature. However, salience feature less considers the motion representations in videos, which is inferior to optical flow feature.
>
>
>
>
>
> > **Q4**: Any idea borrowed from score based/diffusion models for this framework.
>
> **A4**: Thank you for your insightful comment on our framework. We think you provide a novel idea. Trigger-guided generation model may promote the performance of our framework. The generation model could first encode the motion trajectories of each event trigger. Then we could generate any image/video from given event trigger. During the encode/decoder process, the framework may learn more discriminative motion representations.
>
>
>
>
>
>
>
>
>
>
>
> # References
>
> [1] Liu, Liu, et al. "Activity image-to-video retrieval by disentangling appearance and motion." *Proceedings of the AAAI Conference on Artificial Intelligence*. Vol. 35. No. 3. 2021.
>
>
>
> [2] Rajnoha, Martin, Radim Burget, and Lukas Povoda. "Image background noise impact on convolutional neural network training." *2018 10th international congress on ultra modern telecommunications and control systems and workshops (ICUMT)*. IEEE, 2018.
>
>
>
> [3] Zhang, Jingke, et al. "Improved background noise suppression in ultrasound localization microscopy using spatial coherence beamforming." *2021 IEEE International Ultrasonics Symposium (IUS)*. IEEE, 2021.
>
>
>
> [4] Abdessaied, Adnen, Ekta Sood, and Andreas Bulling. "Video Language Co-Attention with Multimodal Fast-Learning Feature Fusion for VideoQA." *Proceedings of the 7th Workshop on Representation Learning for NLP*. 2022.
>
>
>
> [5] Yu, Tiezheng, et al. "Vision Guided Generative Pre-trained Language Models for Multimodal Abstractive Summarization." *Proceedings of the 2021 Conference on Empirical Methods in Natural Language Processing*. 2021.
>
>
>
> [6] Zhou, Huajun, et al. "Interactive two-stream decoder for accurate and fast saliency detection." *Proceedings of the IEEE/CVF conference on computer vision and pattern recognition*. 2020.

---

### Official Review · Reviewer_necu · 2023-08-06

**Soundness:** 4

**Excitement:**

4: Strong: This paper deepens the understanding of some phenomenon or lowers the barriers to an existing research direction.

**Missing References:**

No missing references to my knowledge.

**Paper Topic And Main Contributions:**

This paper presents a novel three stream multimodal event extraction approach that in addition to text and visual appearance features also incorporates video motion features. The Motion feature extraction is combined with a multi level event contrastive module to establish similarity and contrast using motion features. In my view, that is the main innovation of the proposed work. The proposed technique achieves significant improvement over the state of the art and its various aspects are thorough clarified through well designed ablation studies.

**Questions For The Authors:**

1. Most datasets have been set up with no attention to motion. Have you considered creating a dataset that would better show off your algorithm's motion characterization? What would be the challenges there? Note that I am NOT asking you to create a new dataset, only express your educated opinion on a possible new dataset.

**Reasons To Accept:**

1. Thorough literature survey.
2. Innovative incorporation of optical flow features through multi-level event similarity and contrast in a contrastive learning framework.
3. Clear advancement of the state of the art.
4. Insightful ablation studies.

**Reasons To Reject:**

1. I do not see any signficant weaknesses in the paper. The authors do describe their limitations which I don't think count as significant weaknesses.

**Reproducibility:**

5: Could easily reproduce the results.

**Reviewer Confidence:**

5: Positive that my evaluation is correct. I read the paper very carefully and I am very familiar with related work.

**Typos Grammar Style And Presentation Improvements:**

Line 015 typo please replace “moviated” by “motivated” (not good to have typos in the abstract)
LIne 555 please replace “provides” by “provide”

---

> ### Author Rebuttal · Authors · 2023-08-26
>
> We are greatly encouraged by your positive comments. Thanks a lot for all the appreciation. We will revise the typos in our final version.
>
>
> >**Q1**: Most datasets have been set up with no attention to motion. Have you considered creating a dataset that would better show off your algorithm's motion characterization? What would be the challenges there?
>
> **A1**: Thank you for your professional and rigorous thinking. We think it would be promising to set up a dataset with more attention to motion. The new dataset could be constructed jointly with a Computer Vision task, named spatio-temporal action localization. The downstream task of the dataset is to locate the start time and end time of a motion trigger in the text caption from the long video. We suppose the challenge in this task for a model is the capacity of completely understanding the features of text and video. Identifying the start and end time is a high-level ability of recognition motion representation, let alone understanding heterogeneous motion triggers. The dataset needs more refined motion feature extraction, which could better show off our algorithm.

---

### Meta-Review · Area_Chair_RJ3S · 2023-09-18

**Recommendation:** 5

**Metareview:**

The paper proposes a novel method for text-video event extraction based on a three-stream multi-level event contrastive learning approach. In addition to text and visual appearance features, this work incorporates video motion features based on optical flows, significantly improving the proposed system from previous studies.

The reviewers' evaluations are primarily high. I don't find any critical issues in the reviews. "Reasons to reject" from reviewers look limited to minor issues and questions for clarification. The authors responded carefully to such reviewers' concerns and questions in the rebuttal.

One reviewer asked about related works on noise-robust event trigger extraction, the appropriateness of every 16-frame sampling, and the effectiveness of optical flow features against saliency-based features. The authors answered the questions with additional experimental results, showing that the proposed method outperforms different sampling approaches and saliency features.

Another reviewer was concerned about the novelty of optical flow features and motion representation. For these concerns, the authors explained that associating the optical flow features with the event triggers is novel, and the use of motion representation is also novel in text-video event extraction. The reviewer acknowledged the authors' response and raised the soundness score to 4.

Considering the reviews and discussions, the paper is well-written, showing that the proposed method is technically sound and effective for text-video event extraction.

---

### Decision · Program_Chairs · 2023-10-07

**Decision:**

Accept-Main

**Comment:**

The paper proposes a novel method for text-video event extraction based on a three-stream multi-level event contrastive learning approach. In addition to text and visual appearance features, this work incorporates video motion features based on optical flows, significantly improving the proposed system from previous studies.

The reviewers' evaluations are primarily high. I don't find any critical issues in the reviews. "Reasons to reject" from reviewers look limited to minor issues and questions for clarification. The authors responded carefully to such reviewers' concerns and questions in the rebuttal.

One reviewer asked about related works on noise-robust event trigger extraction, the appropriateness of every 16-frame sampling, and the effectiveness of optical flow features against saliency-based features. The authors answered the questions with additional experimental results, showing that the proposed method outperforms different sampling approaches and saliency features.

Another reviewer was concerned about the novelty of optical flow features and motion representation. For these concerns, the authors explained that associating the optical flow features with the event triggers is novel, and the use of motion representation is also novel in text-video event extraction. The reviewer acknowledged the authors' response and raised the soundness score to 4.

Considering the reviews and discussions, the paper is well-written, showing that the proposed method is technically sound and effective for text-video event extraction.